# Genome-Wide Analysis of Flax (*Linum usitatissimum* L.) Growth-Regulating Factor (GRF) Transcription Factors

**DOI:** 10.3390/ijms242317107

**Published:** 2023-12-04

**Authors:** Jianyu Lu, Zhenhui Wang, Jinxi Li, Qian Zhao, Fan Qi, Fu Wang, Chunxiao Xiaoyang, Guofei Tan, Hanlu Wu, Michael K. Deyholos, Ningning Wang, Yingnan Liu, Jian Zhang

**Affiliations:** 1Faculty of Agronomy, Jilin Agricultural University, Changchun 130118, China; lu@mails.jlau.edu.cn (J.L.); wzhjlau@163.com (Z.W.); 20220617@mails.jlau.edu.cn (J.L.); zhaoq217@163.com (Q.Z.); fan711998@163.com (F.Q.); 13722052064@163.com (F.W.); xycxttkx@163.com (C.X.); tagfei@foxmail.com (G.T.); wuhanlu@mails.jlau.edu.cn (H.W.); ningningw@jlau.edu.cn (N.W.); 2Department of Biology, University of British Columbia, Okanagan, Kelowna, BC V5K1K5, Canada; michael.deyholos@ubc.ca; 3Institute of Natural Resources and Ecology, Heilongjiang Academy of Science, Harbin 150040, China

**Keywords:** flax, GRF gene family, phylogeny analysis, fruit development, subcellular localization, expression analysis

## Abstract

Flax is an important cash crop globally with a variety of commercial uses. It has been widely used for fiber, oil, nutrition, feed and in composite materials. Growth regulatory factor (GRF) is a transcription factor family unique to plants, and is involved in regulating many processes of growth and development. Bioinformatics analysis of the GRF family in flax predicted 17 *LuGRF* genes, which all contained the characteristic QLQ and WRC domains. Equally, 15 of 17 *LuGRFs* (88%) are predicted to be regulated by lus-miR396 miRNA. Phylogenetic analysis of GRFs from flax and several other well-characterized species defined five clades; *LuGRF* genes were found in four clades. Most *LuGRF* gene promoters contained cis-regulatory elements known to be responsive to hormones and stress. The chromosomal locations and collinearity of *LuGRF* genes were also analyzed. The three-dimensional structure of *LuGRF* proteins was predicted using homology modeling. The transcript expression data indicated that most *LuGRF* family members were highly expressed in flax fruit and embryos, whereas *LuGRF3*, *LuGRF12* and *LuGRF16* were enriched in response to salt stress. Real-time quantitative fluorescent PCR (qRT-PCR) showed that both *LuGRF1* and *LuGRF11* were up-regulated under ABA and MeJA stimuli, indicating that these genes were involved in defense. *LuGRF1* was demonstrated to be localized to the nucleus as expected for a transcription factor. These results provide a basis for further exploration of the molecular mechanism of *LuGRF* gene function and obtaining improved flax breeding lines.

## 1. Introduction

Growth-regulating factors (GRFs) are small transcription factors that plays an important role in biological processes such as plant growth, development and stress response [1,2]. The N-terminal region of a GRF has two conserved domains, QLQ (Glu, Leu, Glu) and WRC (Trp, Arg, Cys). The QLQ domain interacts with the SYT N-terminal homology domain (SNH) of GRF-interacting factors (GIFs) to form a functional GRF–GIF complex [3]. The amino acid sequence of the WRC domain is more highly conserved than QLQ, containing a nuclear localization signal domain and a C3H-type zinc finger domain, which can be combined with the cis-acting elements of downstream genes to regulate the temporal and spatial expression of genes [4]. Compared with the N-terminus, the C-terminus of the GRF protein varies widely: motifs such as TQL (Thr, Gln, Leu), GGPL (Gly, Gly, Pro, Leu) and FFD (Phe, Phe, Asp) often appear in the C-terminal end of the protein, although these are not highly conserved [5,6]. In addition, most GRF genes described to date are targets of miR396, and post-transcriptional regulation of miR396/GRFs is prevalent in plants [7,8,9].

The first GRF gene cloned was *OsGRF1* from rice (*Oryza sativa*), which is induced by gibberellin in the internode meristem, thus promoting stem elongation [10]. GRF genes are generally members of small gene families within each species. With the completion of sequencing for many plant genomes, the GRF gene family has been identified in many plants, including *Arabidopsis thaliana* [5], *Glycine max* [11], *Ananas comosus* [12], *Solanum lycopersicum* [13], *Nelumbo nucifera* [14] and *Panax ginseng* [15]. However, the identification of GRF genes in flax has not been studied.

Different members of the GRF family show different expression patterns in different parts of the plant. Generally speaking, the expression levels of GRFs are higher in the tissues or organs that have active growth, and gradually decrease during the maturation of the organs [5,7]. Some GRF genes are also responsive to stress in various tissues and organs [16,17]. For example, *AtGRF7* can inhibit *Dehydration Response Element Binding 2A* (*DREB2A*) to regulate plant responses to osmotic stress [2]. Within a species, different GRF members show both functional redundancy and functional differentiation [1]. In *Arabidopsis thaliana* transgenic lines overexpressing *AtGRF1* or *AtGRF2*, the cotyledon and leaf sizes increase significantly due to an increase in cell size, while the overexpression of *AtGRF5* increases the leaf area by increasing the number of cells. Thus, GRF transcription factors can regulate organ size by regulating cell expansion or proliferation [5,18]. Compared with the wild type, the cotyledon and rosette leaf areas of *Arabidopsis* triple mutants *grf1*/*grf2*/*grf3* were significantly reduced, while the single-gene mutants showed no significant phenotypic differences, indicating that these genes have functional redundancy [19]. *AtGRF9* negatively regulates the leaf size by inhibiting cell proliferation [20]. Overexpression of *OsGRF4* in rice will increase the grain width, grain length and ear length related to rice yield [21,22,23], while the inhibition of *OsGRF3*, *OsGRF4* and *OsGRF5* expression will shorten rice plants and delay the inflorescence development time [24]. *OsGRF1* and *OsGIF1* coordinate to regulate the internal and external conditions of rice leaf growth, thus affecting leaf growth [25]. In *Gossypium herbaceum*, the expression of *GhGRF3*, *GhGRF4*, *GhGRF5*, *GhGRF7* and *GhGRF16* decreased under salt stress treatment, suggesting that they play a regulatory role in growth and development under abiotic stress conditions [26].

Flax (*Linum usitatissimum* L.) is an ancient bast fiber crop and oil crop and is widely cultivated especially in China, Canada, France, Russia, Ukraine, the United States, Argentina, etc. [27]. According to their intended end use, flax varieties can be classified into three types: oil, fiber and dual-purpose [28,29]. Flaxseed (linseed) is rich in lignans, dietary fiber, protein and other nutrients, including α-linolenic acid, which at 55% by weight is one of the best sources of ω-3 fatty acids for human nutrition [30]. Until now, there have been no reports about the GRF genes in flax. In this study, flax GRF family genes were systematically surveyed and their physicochemical properties, sequence characteristics, phylogeny, promoter cis-acting elements and gene expression patterns were analyzed, which provided the basis for further study of biological functions of flax GRF genes. This is the first comprehensive study reported on flax GRFs.

## 2. Results

### 2.1. Identification and Phylogenetic Analysis of Flax GRF Gene Family Members

A total of 17 GRF family genes were identified in the genome of flax (var. Longya10). This was accomplished via the alignment of predicted flax genes with GRF genes from Arabidopsis thaliana. All 17 predicted LuGRF proteins were confirmed to contain the WRC (PF08879) and QLQ (PF08880) domains. These genes were named *LuGRF1*-*LuGRF17* according to their chromosomal positions. In terms of physical and chemical properties, the longest proteins were LuGRF2 and LuGRF11, which each contained 496 amino acids, and the smallest protein was LuGRF8, which contained 151 amino acids. The molecular weight of LuGRF protein was predicted to be between 17.64 and 55.7 kDa. Of the proteins encoded by the *LuGRF* gene family, 88% had a pI greater than 7, indicating that most of the proteins encoded by the flax GRF gene family were rich in basic amino acids. The instability index ranged from 50.02 to 77.55, indicating that GRFs are probably unstable proteins. The aliphatic index of LuGRF proteins ranged from 49.57 to 75.16. Hydropathicity analysis showed that all LuGRF proteins were hydrophilic proteins. Subcellular localization predicted that all *LuGRF* genes were expressed in the nucleus (Table 1).

To better understand the evolutionary relationship between GRF genes in different species, the maximum likelihood (ML) method was used to construct a phylogenetic tree of 51 GRF genes from Arabidopsis thaliana (9), Zea mays (16), Oryza sativa (9) and flax (17) (Figure 1). The results showed that the 51 GRF genes could be divided into five subfamilies, which were named I–V. Subfamilies I through IV contained GRF clusters of monocotyledonous and dicotyledonous plants, while subfamily V contained GRF clusters of monocotyledonous plants only. The 17 *LuGRF* genes were distributed in every subfamily except subfamily V. The most numerous genes were in subfamily IV, which included five members: *LuGRF3*, *LuGRF4*, *LuGRF12*, *LuGRF14* and *LuGRF15*. At the same time, it was found that all flax GRF gene families are closely associated with Arabidopsis, which proved that the flax and Arabidopsis GRF families were the most closely related among these five species.

### 2.2. Analysis of LuGRF Gene Structure and Conserved Domain

Analysis of the *LuGRF* gene structures showed the number of exons was between two and six, among which *LuGRF13* and *LuGRF2* contained six and five exons, respectively; *LuGRF8*, *LuGRF15* and *LuGRF15* all contained three exons; both *LuGRF3* and *LuGRF12* had two exons and the rest of the *LuGRF* gene family had four exons (Figure 2D). The exon and intron structure was generally conserved within each of the subgroups I–IV. Analysis of the conserved domains of flax GRF genes showed that the WRC and QLQ conserved domains were found in all flax GRF family members (Figure 2C). A total of 10 conserved motifs of those predicted using MEME were found within flax GRFs (Figure 2B). The results showed that all *LuGRFs* contained motif 1, 2 and 3; 14 GRFs contained motif 4; and 8 GRFs contained motif 9, which constituted the typical conserved regions of the *LuGRF* gene family. Motif 5 and motif 7 were specific to their GRF clades.

### 2.3. Chromosomal Localization and Collinearity Analysis of LuGRF Gene Family

The 17 flax GRF genes were unevenly distributed in 10 out of 15 flax chromosomes (Figure 2E). Three of the genes were distributed on chromosome 3; chromosomes 1, 5, 7 and 10 contained two *LuGRF* genes and chromosomes 4, 6, 9 and 14 all contained one *LuGRF* gene. Gene duplication often results in genetic mutations that lead to new functions and play a key role in plant adaptation. To study gene duplication events in the history of the *LuGRF* family, TBtools was used to construct Circos maps (Figure 3A). The map shows five duplicate gene pairs within the *LuGRF* gene family: *LuGRF1*-*LuGRF10*; *LuGRF3*-*LuGRF12*; *LuGRF4*-*LuGRF14*; *LuGRF5*-*LuGRF13*; *LuGRF14*-*LuGRF15*. These gene pairs may be related to flax WGD (Whole-Genome Duplication) events.

The evolutionary relationship between flax and *Arabidopsis* was also further analyzed, and nine colinear genes were identified in the colinear region of the genome (Figure 3B). The collinearity of chromosomes 2, 4 and 7 of flax was at most 2, and the *LuGRF* gene was colinear with chromosomes 2, 3 and 4 of *Arabidopsis*, respectively.

### 2.4. Cis-Acting Elements and miRNA Prediction

We predicted the cis-acting elements 2000 bp upstream of each GRF promoter region in flax and plotted them by removing common elements such as TATA-box and CAAT-box (Figure 4A). The analysis identified 371 cis-acting elements, which could be divided into four categories: development-related elements; environmental-stress-related elements; hormone-responsive elements and light-responsive elements (Figure 4B,C). The major hormone-responsive elements were abscisic-acid-responsive element (ABRE); auxin-responsive element (AuxRR-core and TGA-element); MeJA-responsive element (TGACG-motif and CGTCA-motif); gibberellin-responsive element (P-box and GARE-motif) and salicylic-acid-responsive element (TCA-element). The MeJA-responsive element was the most abundant hormone-related cis-element in the upstream regions of the *LuGRF* gene family. The second largest category was light-responsive elements, including 131 components such as GT1-motif, G-box and LAMP-element (35.3%). The third category was environmental-stress-related elements with a total of 75 components (20.22%), including an anaerobic induction element (ARE), defense- and stress-responsive element (TC-rich repeats), low-temperature responsive element (LTR), MYB binding site involved in drought inducibility (MBS) and MYB binding site involved in light responsiveness (MRE). The fourth category was development-related elements with 29 components (7.81%), including an endosperm expression element (GCN4_motif), meristem expression element (CAT-box) and MYBHv1 binding site (CCAAT-box).

The results of MiRNA target prediction showed that there were 26 miRNA targets in 15 *LuGRF* genes except *LuGRF3* and *LuGRF12* (Table 2). Among all *LuGRF* genes, *LuGRF14* had the most miRNA targets at 14. We found that different *LuGRF* genes could be targeted by the same miRNA in the same subfamily. For example, in subfamily I, *LuGRF5*, *LuGRF6*, *LuGRF13* and *LuGRF17* could be simultaneously targeted by lus-miR396, and they could also be targeted by different miRNAs in the same subfamily. For example, *LuGRF6* could be targeted by either lus-miR396 or lus-miR168. Analysis of the cleavage site of miRNA396 showed that miRNA396a and miRNA396c and miRNA396b and miRNA396e had the same cleavage site, and both cut the WRC conserved domain of the *LuGRF* gene. We also found that only *LuGRF8* had two cleavage sites and cut the CDS region (Appendix A). The results showed that lus-miR396 is the main miRNA target of the GRF gene family.

### 2.5. Structural Characterization and Subcellular Localization of LuGRF Protein

The secondary structure predictions shows that the LuGRF protein is composed of an alpha helix, an extended strand and a random coil (Table 3). Random coil amino acids accounted for the highest proportion (47.16–70.87%). This was followed by extended strand amino acids (15.12–25%) and alpha helix amino acids (9.09–31%). Further, 3D protein structure prediction found that all *LuGRF* family members except *LuGRF8* had a similar protein structure (Appendix A), demonstrating that they had similar protein functions.

The highly expressed gene *LuGRF1* was selected for subcellular localization, and the strains of *Agrobacterium tumefaciens* EAH105 containing pGD:LuGRF1-RFP and pGD:RFP (control) were inoculated into young tobacco epidermises for transient expression. The results showed that pGD:LuGRF1-RFP was expressed in the nucleus (Figure 5), which was consistent with the predicted results (Table 1). The pGD:RFP was localized both inside and outside of the nucleus.

### 2.6. GO Enrichment Analysis of LuGRF

GO enrichment analysis of *LuGRF* genes in flax showed that *LuGRF* gene plays a role in molecular function, cell composition and biological processes (Figure 6A). Among them, the main participants include negative regulation of cell population proliferation, plant organ development, system development cell population proliferation, leaf development, regulation of post-embryonic development and root development and other biological processes.

### 2.7. Analysis of RNA-Seq Data and LuGRF Gene Expression Profile

To investigate the potential functions of the *LuGRF* gene, we analyzed the expression in publicly available RNA-seq data, and analyzed the expression patterns of flax roots, seeds, stems, embryos, anthers, ovaries, fruits, pistils and stamens in different tissues. The expression of related *LuGRF* genes was mapped (Figure 6C). The results showed that the expression of the *LuGRF* gene family was different in different tissues, and most of the family members were expressed in flax fruit. These members are expressed at a low level in the roots and stems, and only *LuGRF3, LuGRF16* and *LuGRF17* were expressed in the seeds and stamens. Two members (*LuGRF6* and *LuGRF8*) were highly expressed in the heart-stage embryos, while the rest had low or no expression. Three members (*LuGRF1*, *LuGRF6* and *LuGRF8*) were highly expressed in the globular-stage embryos. *LuGRF5*, *LuGRF13* and *LuGRF16* were highly expressed in cotyledon embryos and anthers, and the expression of *LuGRF16* was the highest in mature embryos. Only *LuGRF4*, *LuGRF7*, *LuGRF9* and *LuGRF14* were highly expressed in the ovaries, but not in other members. Therefore, *LuGRF* genes may play a regulatory role in flax fruit and flowering development.

Published abiotic RNA-seq data were processed to further study the expression matrix of *LuGRF* genes in stress response (Figure 6B). The expression of *LuGRF* genes was relatively high in the root tissues without salt treatment, while the expression was significantly inhibited in the root tissues after salt stress. Compared with the control group, three genes (*LuGRF3*, *LuGRF12* and *LuGRF16*) of the flax shoot significantly increased in response to salt stress. When the flax stem was subjected to heat stress, compared with the control group, the expression of eight *LuGRF* genes was found to be increased, among which the expressions of *LuGRF4*, *LuGRF13* and *LuGRF14* were significantly increased.

### 2.8. Plant Hormone Stress and qRT-PCR

To verify whether the *LuGRF* genes play a role in the response of the ABA and MeJA plant hormones, the transcript abundance of 17 *LuGRF* genes in leaf tissues were measured using qRT-PCR (Figure 7). The gene expression in the leaves was detected at 0 h, 4 h, 8 h, 12 h and 24 h, and the expression levels of 0 h samples were compared (Figure 8). The expression levels of *LuGRF1* and *LuGRF7* were also significantly up-regulated with the increase in ABA stress time. The expression levels of *LuGRF11* reached an extreme value at 4 and 8 h of ABA stress, which increased by about 3.3 times compared with the control. The expression levels of *LuGRF12* were the highest at 8 h of ABA stress, which increased by 2.3 times compared with the control. We also found that four genes (*LuGRF3*, *LuGRF8*, *LuGRF9*, *LuGRF16*) were significantly down-regulated under ABA stress, with the expression of *LuGRF3* reaching its maximum value at a stress of 8 h and *LuGRF8* reaching its maximum value at a stress of 12 h. The expression levels of *LuGRF9* and *LuGRF16* were the lowest after a stress of 24 h, which may be related to the negative regulation of ABA.

MeJA treatment significantly induced significantly up-regulated expression of two genes (*LuGRF1*, *LuGRF11*). The transcription level of *LuGRF1* increased by 4.2 times at 12 h, and that of *LuGRF11* increased by 2.2 times at 24 h. We also found that five genes were down-regulated, namely *LuGRF2*, *LuGRF3*, *LuGRF6*, *LuGRF15* and *LuGRF16*, whose transcription levels were down-regulated but not significant compared with the control after treatment for 24 h.

## 3. Discussion

GRFs are plant-specific transcription factors that not only play an important role in regulating various plant growth and development processes, but also regulate many aspects of plant stress responses [1,19,31]. However, until now, there have been no reports about the GRF gene family in flax so far. In this study, 17 *LuGRF* genes were identified based on their similarity to *Arabidopsis* GRF proteins and the presence of the QLQ and WRC domains. The number of flax GRF genes (17) is more than *Arabidopsis* (9) or pineapple (8), and similar to the number identified in ginseng (20) and maize (17). This suggests that GRF genes may have undergone sustained family-specific amplification during evolution [5,12,15,32].

Phylogenetic analysis of GRFs from diverse species identified five subfamilies (I–V), and the *LuGRF* genes were restricted to four of these subfamilies (I–IV) (Figure 1). Members within the same subfamily may have similar functions [33]. For example, overexpression of *AtGRF3* increases organ size, and it is speculated that *LuGRF6* and *LuGRF7* are also involved in organ development [34]. Previous studies have found that overexpression of *AtGRF5* and *AtGRF6* has a positive effect on the proliferation of transgenic rapeseed (*Brassica napus*) callus cells, suggesting that *LuGRF15* may play an important regulatory role in cell proliferation and differentiation in the leaf primordium of flax cells [35].

Phylogenetic analysis showed that *LuGRF* genes were clustered in four clades (Figure 2A). Based on motif analysis, motif 1 (WRC) and motif 2 (QLQ) were found in all LuGRF proteins (Figure 2C), and there are different conserved protein motifs in LuGRF proteins. The differences between these subfamilies demonstrate the functional diversity of GRF members [4,5,32]. As shown in Figure 2B, WRC is closely related to other motifs, and it is possible that this domain plays a role in DNA binding. The C-terminal motif of GRF protein plays an important role in different plant tissues and organs [5,15,32]. Flax GRF C-terminal protein can bind to cis-acting elements of downstream genes to regulate the temporal and spatial expression of genes [10]. The N-terminal and C-terminal domains of flax GRF protein participate in plant growth and development [4].

The collinearity analysis of *LuGRF* showed that there were five gene pairs with repeated fragments (*LuGRF1*-*LuGRF10*, *LuGRF3*-*LuGRF12*, *LuGRF4*-*LuGRF14*, *LuGRF5*-*LuGRF13*, *LuGRF14*-*LuGRF15*) (Figure 3A). There were also colinear relationships with nine genes of *Arabidopsis thaliana* (Figure 3B) [14,15,36].

MicroRNA is involved in the regulation of GRF gene expression and is an important factor that negatively regulates GRF gene expression [7]. Previous studies have found that the miR396-GRF module is crucial for controlling the length of rice leaves, and the single-strand DNA corresponding to the sequence of miR396e can effectively down-regulate the expression of GRF in developing leaves, resulting in the shortening of the length of leaves [9]. Inhibition of the expression of Sbi-miR396d/e may increase the expression of SbiGRF1/5/8, affecting the flower organs and seed development of *Sorghum bicolor* [37]. Overexpression of microRNA396a in *Arabidopsis* can significantly reduce the expression level of miRNA396 target genes (*AtGRF1*, *AtGRF2*, *AtGRF3*, *AtGRF4*, *AtGRF7*, *AtGRF8*, *AtGRF9*) and lead to abnormal pistils [8]. miR396 may induce somatic embryo development by controlling tissue sensitivity to auxin treatment [38]. In this study, 15 *LuGRFs* (88%) were regulated by lus-miR396 (Table 2), suggesting that lus-miR396 may be an important factor negatively regulating GRF gene expression in flax and play an important role in the growth and development of flax (Appendix A).

The 3D structure of GRF protein in the same clade is conserved, and its gene structure, conserved domain and expression pattern are consistent (Appendix A). Using model structure prediction analysis, we can deepen our understanding of the biological functions of GRF family members. We constructed a fusion expression vector of the *LuGRF1* gene and transferred the gene into tobacco leaves, and found that the gene was expressed in the nucleus (Figure 5), which was consistent with the results reported in the literature [14].

Studies have shown that GRFs play a crucial role in organ development [39,40], including expression in different parts of the roots, stem apex meristem and flowers, especially in cell proliferation and differentiation [4,5,41]. According to RNA-seq data analysis, *LuGRF* expression is different in different tissues, but most *LuGRF* genes are highly expressed in embryos, during flower development and in fruit, and cell proliferation is vigorous (Figure 6C). Studies have shown that overexpression of ath-miR396 is characterized by small and narrow leaves and embryonic defects, and that it plays a role in flower organ development [42], and the overexpression of NTGRF-like genes regulates leaf and flower development in tobacco [43]. GRF genes are strongly expressed in the young leaves, flower buds and bud tips of rice, which affects plant growth and development by regulating cell proliferation [4]. The *OsGRF1* gene not only regulates leaf shape but also flower formation, and *OsGRF6* also plays an important role in the development of flower organs [44,45]. There are few reports on the correlation between GRF and fruit development, but some studies have found that when the expression of miRNA396a/b in transgenic tomatoes is down-regulated, the expression level of its target gene GRF is up-regulated, and correspondingly, the flowers, calyx and fruits of transgenic tomato plants become significantly larger [46]. It was further confirmed that *LuGRF* genes may play an important role in regulating flower organ development, but their function in fruits and embryos needs to be further verified.

Plants have evolved a range of stress resistance mechanisms, in which transcription factors play a key regulatory role. GRF-like transcription factors play an important role in coordinating stress responses and defense signals during plant growth and development [45,47,48]. For example, under stress conditions, overexpression of *Populus davidiana* × *P*. *bolleana PdbGRF1* enhanced tolerance to salt stress [49]. In strawberries, *FvGRP4* and *FvGRP9* are up-regulated under high temperature stress [50]. Transcriptome data showed that three *LuGRF* genes were up-regulated after salt treatment, and eight *LuGRF* genes were found to have increased expression after heat treatment (Figure 6B). The response of *LuGRF* genes to heat stress was relatively strong under the two abiotic stresses, and *LuGRF* genes may be involved in biological processes related to abiotic stress response. The response and function of *LuGRF* genes to environmental stress in flax need to be further verified.

Cis-element analysis of the promoter region of *LuGRF* genes showed a large number of development-related elements, environmental-stress-related elements, hormone-responsive elements and light-responsive elements (Figure 4A), most of which are related to hormone response (Figure 4B,C), including abscisic acid and MeJA response elements. These results suggested that the expression level of *LuGRF* might be induced by ABA and MeJA to some extent. In this study, qRT-PCR was used to analyze the expression patterns of flax in different tissues and treatments (Figure 7). Four genes were significantly upregulated by ABA treatment and two genes were significantly upregulated by MeJA treatment (Figure 8). GRF transcription factor was first found to regulate plant growth via the gibberellin pathway, and exogenous GA treatment of rice could increase the transcription levels of *OsGRF1*, *OsGRF2*, *OsGRF3*, *OsGRF7*, *OsGRF8* and *OsGRF10* 4–10 times over [4]. Six GRF genes were identified in the transcriptome of tea tree, and it was found that the expression level of three members was increased after ABA treatment, and the expression level of five members was decreased after MeJA treatment [51]. Our study found that *LuGRF1*, *LuGRF7*, *LuGRF11* and *LuGRF12* expression was induced by exogenous ABA, consistent with the role of *LuGRF* genes in abiotic stress tolerance [52]. Jasmonic acid plays a key regulatory role in plant growth and development and various physiological processes, including organ formation, the reproductive process, fruit maturation and senescence and resistance to biotic and abiotic stresses [53,54]. In our study, we found that the expression levels of 15 *LuGRF* genes were weakly affected by MeJA, and only the transcription levels of *LuGRF1* and *LuGRF11* increased significantly at 12 h and 24 h after treatment, suggesting that *LuGRF1* and *LuGRF11* may be involved in embryo development via the signaling pathway of MeJA. Moreover, putative cis-regulatory elements associated with hormone response, such as ABA and MeJA, were recognized in the promoter region of *LuGRF* genes. In plant species, hormones such as ABA and MeJA are more linked with stress cell signaling pathways [55,56]. In addition, several transcription factors and ion transporter genes are induced by ABA and MeJA signaling in response to environment stimuli [57,58]. Our results suggest that *LuGRF* genes might be directly induced by ABA and MeJA. However, there are few studies on GRF genes and MeJA hormone treatment, and the response to and function of *LuGRF* in flax hormone stress need to be further verified.

## 4. Materials and Methods

### 4.1. Plant Material

The flax (*Linum usitatissimum*) variety Longya10 was selected as the focus of this study. The flax seeds were disinfected with 75% alcohol for 10 min, washed with sterile water, planted on soil and transferred to a greenhouse under 26/18 °C 16/8 h light. When flax seedlings grew to 6–7 cm tall, the treatment group was sprayed with ABA (80 mM), MeJA (80 mM) or distilled water. At 0, 4, 8, 16 and 24 h after spraying, the leaves were sampled, and this was repeated three times for each treatment group. All the samples were frozen with liquid nitrogen and stored in an ultra-low temperature refrigerator at −80 °C for RNA extraction.

### 4.2. Identification of LuGRF Gene Family in Flax

The flax genome sequence was downloaded from NCBI (entry number QMEI02000000) and the other genome annotation files were obtained at https://figshare.com/articles/dataset/Annotation_files_for_Longya-10_genome/13614311 (accessed on 1 August 2023). We used two methods to identify the flax GRF genes. First, *Arabidopsis* GRF protein sequences were downloaded from the TAIR website (https://www.arabidopsis.org/, accessed on 15 August 2023) and these were used as queries in a BLASTP search of flax’s predicted proteins. We also obtained profile-hidden Markov models of the QLQ (PF08880) and WRC (PF08879) domains from the Pfam database (http://pfam.xfam.org/, accessed on 15 August 2023), and used them to search for the predicted flax proteins. Flax proteins that aligned with *Arabidopsis* GRF proteins, but that did not contain the QLQ and WRC domains, were discarded. The remaining proteins were defined as flax LuGRF proteins, resulting in 17 flax GRF genes. Using the ExPASy ProtParam tool (https://web.expasy.org/protparam/, accessed on 20 August 2023), analysis of the physical and chemical properties, including the coding length, number of amino acids, molecular weight (MW) and theoretical isoelectric point (pI), of the *LuGRFs* was calculated. Subcellular localization prediction was performed using the BUSCA website (http://www.busca.cn, accessed on 20 August 2023). Using the MBC website (http://cello.life.nctu.edu.tw, accessed on 20 August 2023), we also predicted the signal peptides and transmembrane domains.

### 4.3. Phylogenetic Analysis

The Clustal W tool in MEGA (version 11) was used to compare the full-length amino acid sequences of the GRF proteins from *Oryza sativa*, *Zea mays*, *Arabidopsis thaliana* and *Linum usitatissimum* using the default parameters. Using the MEGA version 11 software, the neighbor joining algorithm and default settings (neighbor-Joining; Parameter: Bootstrap 1000) were used to construct a phylogenetic tree under the maximum likelihood (ML) principle. Then, we visualized the constructed tree using iTOL version 6 (https://itol.embl.de/, accessed on 2 September 2023).

### 4.4. Chromosome Localization, Conserved Domain and Conserved Motif of LuGRF

Information on the location of *LuGRF* on the chromosomes was obtained from the flax genome FASTA file and the gff3 file. Conservative structure domain identification on the NCBI CD-search website (https://www.ncbi.nlm.nih.gov/Structure/BWRPSB/BWRPSB.Cgi, accessed on 2 September 2023) and the use of MEME (http://alternate.meme-suite.org/tools/meme, accessed on 2 September 2023) was involved in analyzing the conserved motifs of the LuGRF proteins, and we visualized them using TBtools version 2.026 [59].

### 4.5. Cis-Acting Elements and miRNA Prediction

The genomic sequence from 2000 bp upstream of each *LuGRF* gene was extracted using TBtools. Then, the online PlantCARE website (http://bioinformatics.psb.ugent.be/webtools/plantcare/html/, accessed on 5 September 2023) was used to predict the cis-regulatory elements in the upstream regions using TBtools for visualization of the final result.

The miRNA targeting *LuGRF* was predicted using the psRNATarget website (https://www.zhaolab.org/psRNATarget/analysis?function=3, accessed on 5 September 2023) with the default parameters [60].

### 4.6. Three-Dimensional Protein Structure Prediction and GO Enrichment Analysis

First, the protein secondary structure of the LuGRF proteins was predicted using GOR IV (https://npsa-prabi.ibcp.fr/cgi-bin/npsa_automat.pl?page=/NPSA/npsa_gor4.html, accessed on 8 September 2023). Then, the 3D structure of the protein was predicted using the SWISS-MODEL online tool (https://swissmodel.expasy.org/, accessed on 7 September 2023).

First, we downloaded the go-base.obo file in TBtools. Then, we input the protein information of the entire flax genome into the online website eggNOG-mapping (http://eggnog-mapper.embl.de/, accessed on 7 September 2023) to obtain the annotation files. Finally, the GO Enrichment function in TBtools was used for visualization.

### 4.7. Subcellular Localization

RNA was extracted from young flax leaves and reverse-transcribed into cDNA and used as a PCR template to clone the full-length coding region of *LuGRF1*. The primers were designed (Appendix A) using Primer 5 to allow ligation to the pGM-T vector and transformation into competent cells of *E. coli*. The resulting pGM-T-GRF1 was digested using XhoI and SalI enzymes and subcloned into the same sites in pGD-RFP. After the *LuGRF1* fragment was ligated into pGD-RFP, it was transferred into *E. coli*, the recombinant plasmid pGD-GRF1-RFP was extracted, and the localization vector was successfully constructed. The pGD-RFP and pGD-GRF1-RFP plasmids were transferred into *Agrobacterium tumefaciens* GAH105 receptor cells; then, the two bacterial liquids were expanded and the bacterial weight was suspended in the infection buffer, incubated at 28 °C for 3 h, injected into tobacco leaves with a syringe, and the location was observed under confocal microscopy after 3 days of incubation in the dark.

### 4.8. Analysis of RNA-Seq Data and Expression Patterns of LuGRF

Four flax RNA-seq data sets were used in this study: (i) pistils, stamens, fruits and shoots (NCBI SRA PRJNA1002756) (https://www.ncbi.nlm.nih.gov/sra/?term=, accessed on 8 September 2023); (ii) torpedo_embryo, root, ovary, mature_embryo, heart_embryo, globular_embryo, cotylden_embryo, anther, seeds (PRJNA663265); (iii) roots and leaves after salt stress (PRJNA977728) [61]; (iv) after thermal stress (PRJNA874329). Data were processed using the fastp [62] filter connector with hisat2 [63] to map to the Longya10 reference genome, using the tidyverse [64], Rsubread [65], limma [66] and edgeR [67] R packages for further quantification. Finally, TBtools was used to draw a heat map of the log_2_ values of FPKM.

### 4.9. RNA Extraction, and qRT-PCR

The RNA was extracted using an RNA extraction kit (BIOMGA, San Diego, CA, USA), and after NanoDrop-2000 concentration detection, the RNA was reverse-transcribed into cDNA. qRT-PCR was performed using the TB Green^TM^ Premix ExTaq^TM^ II (TaKaRa Bio, Kyoto, Japan) fluorescence quantitative kit. The PCR reaction system was 20 μL: 2× mix 10 μL; upstream and downstream primers (Appendix A) were 1 μL each; cDNA template 2 μL; RNase free water 6 μL. The qRT-PCR procedure was performed at 50 °C for 2 min; 95 °C for 10 min; 95 °C 15 s; 60 °C 15 s; 72 °C 15 s; 40 cycles. Each sample was repeated 3 times, GAPDH was selected as the reference gene and the relative gene expression was calculated using the 2^−ΔΔCt^ method.

## 5. Conclusions

This study provides a comprehensive and systematic analysis of the flax GRF transcription factor family for the first time. A total of 17 members of the GRF transcription factor family have been widely identified in the flax genome. The phylogenetic relationships, gene structure, conserved motifs and collinearity were analyzed using bioinformatics analysis methods. The three-dimensional structure prediction of the protein revealed that *LuGRF* has a similar structure and may have similar functions. MicroRNA prediction found that lus-miR396 is the main miRNA target of the *LuGRF* gene family. Most *LuGRF* gene promoters contain cis-regulatory elements responding to hormones and stress. Based on the expression profiles and qRT-PCR, most *LuGRF* genes were associated with development (flax fruit and embryo, etc.), hormones (MeJA, ABA, etc.) and abiotic stresses (salt, heat, etc.). We also verified the expression of *LuGRF1* genes in the nucleus using subcellular mapping. These analysis results help us understand the role of flax GRF genes in development and environmental stress conditions, and provide a basis for studying the molecular mechanisms of flax GRF transcription factors.

## Figures and Tables

**Figure 1 ijms-24-17107-f001:**
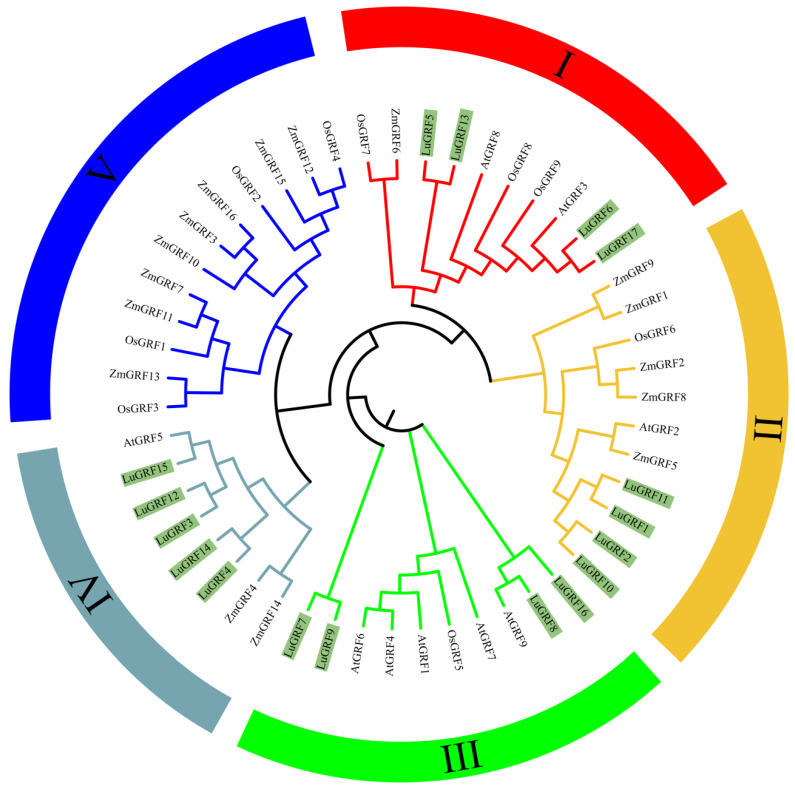
GRF protein phylogenetic tree. Lu represents *flax* genes, At represents *A. thaliana* genes, Zm represents maize genes and Os represents rice genes. Red represents subfamily I, yellow represents subfamily II, green represents subfamily III, blue–green represents subfamily IV and blue represents subfamily V. Dark green marks all flax GRF genes.

**Figure 2 ijms-24-17107-f002:**
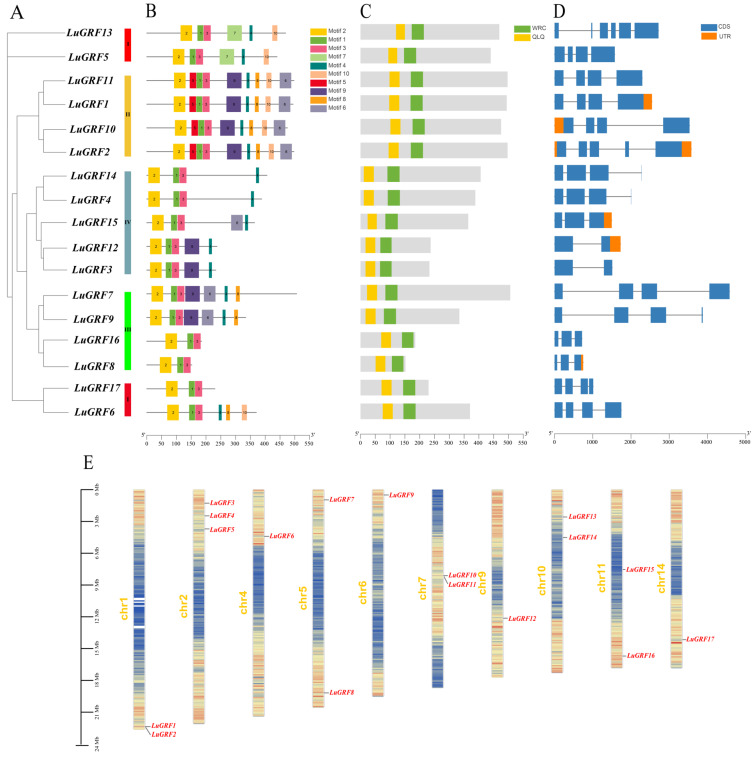
Structure, conserved domain, motif and chromosome distribution of *LuGRF* genes. (**A**) Phylogenetic tree of *LuGRF* genes. (**B**) Motif analysis of protein encoded by *LuGRF* genes. Gray represents the total length of *GRF* genes. (**C**) Conserved domain of *LuGRF* gene. (**D**) *LuGRF* gene structure. (**E**) Chromosome distribution of *LuGRF* gene. Set the sliding window size to 100 kb, with red to blue representing the gene density from high to low. Note: CDS, CodingSequence; UTR, Untranslated Region.

**Figure 3 ijms-24-17107-f003:**
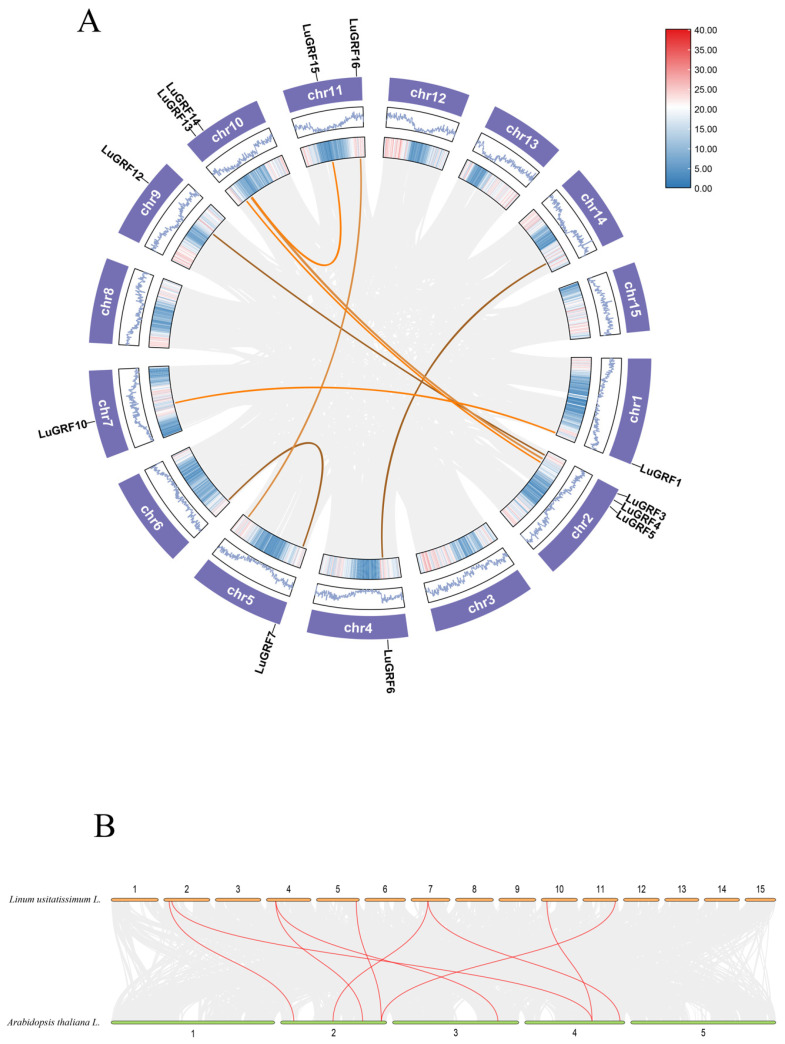
*LuGRF* collinearity analysis. (**A**) Analysis of collinearity between *LuGRF* genes. (**B**) Collinearity of GRF gene in flax and *A. thaliana*. The gray lines represent all collinear gene pairs, and the colored lines represent collinear *LuGRF* gene pairs.

**Figure 4 ijms-24-17107-f004:**
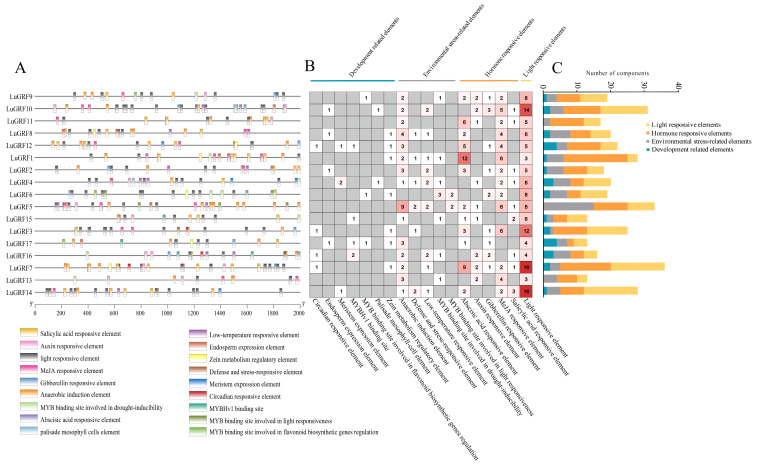
Analysis of cis-regulatory elements of flax *LuGRF* gene. (**A**) Distribution of cis-acting elements in the promoter region of *LuGRF* gene. (**B**,**C**) *LuGRF* gene promoter number statistics. Yellow represents light response elements, orange represents hormone response elements, gray represents environmental stress related elements and blue represents development related elements.

**Figure 5 ijms-24-17107-f005:**
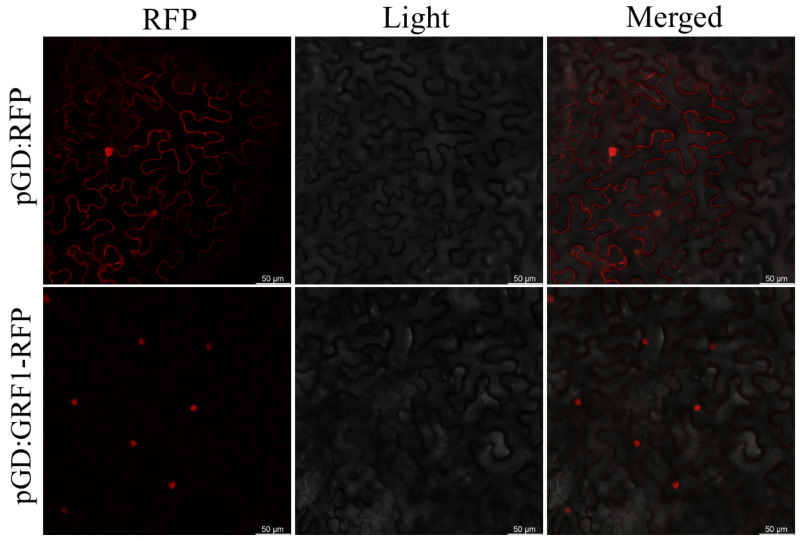
Subcellular localization analysis of LuGRF1 protein in tobacco cells. The scale bar is 50 µm.

**Figure 6 ijms-24-17107-f006:**
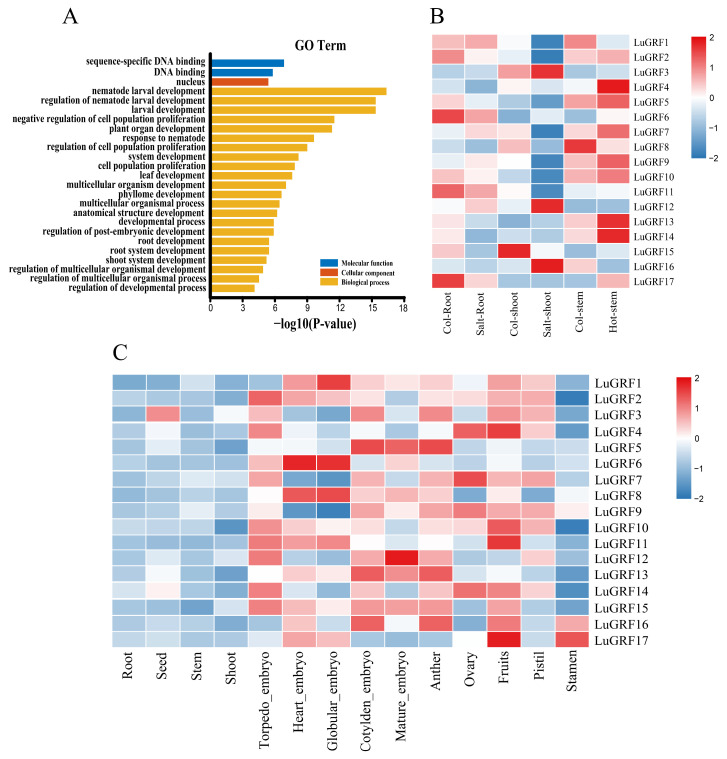
The expression patterns of *LuGRF* genes and GO enrichment were analyzed using RNA-seq data. (**A**) GO enrichment analysis of *LuGRF* genes. (**B**) Expression pattern of *LuGRF* genes after salt treatment and heat treatment. Col stands for control tissue. Salt stands for salt treatment and Hot stands for heat treatment. (**C**) The expression level of *LuGRF* in different tissues of flax.

**Figure 7 ijms-24-17107-f007:**
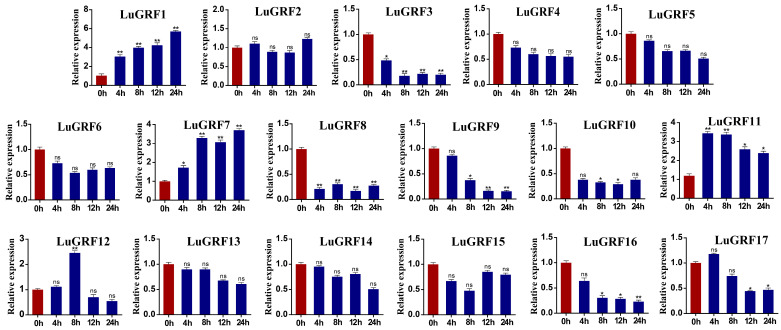
Analysis of expression pattern of *LuGRF* gene in ABA-hormone-induced response. The red column represents the control group and the blue column represents the treatment group. Using Student’s *t*-test, asterisks indicate statistically significant differences (* *p* < 0.05; ** *p* < 0.01). ns represents no significance. Data are shown as mean ± SD from three independent experiments.

**Figure 8 ijms-24-17107-f008:**
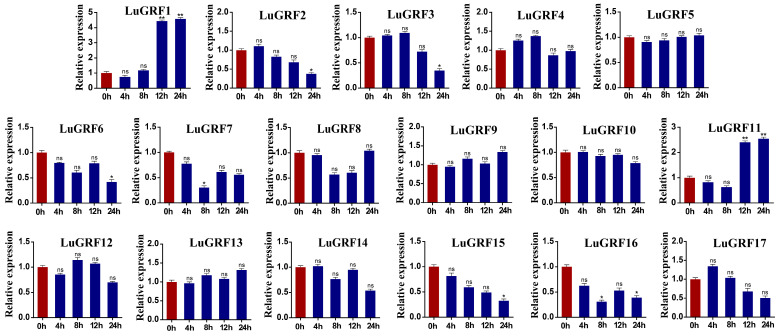
Analysis of expression pattern of *LuGRF* genes in MeJA-hormone-induced response. The red column represents the control group and the blue column represents the treatment group. Using Student’s *t*-test, asterisks indicate statistically significant differences (* *p* < 0.05; ** *p* < 0.01). ns represents no significance. Data are shown as mean ± SD from three independent experiments.

**Table 1 ijms-24-17107-t001:** Characteristics of GRF genes in flax.

Gene	Gene ID in Genome	Chromosome Location (bp)	Number of Amino Acids	Molecular Weight (KDa)	PI	Instability Index	Aliphatic Index	Grand Average of Hydropathicity (GRAVY)	Subcellular Localization
*LuGRF1*	L.us.o.m.scaffold210.24	Chr01:22356867-22359416(+)	493	52,225.65	8.28	59.68	59.05	−0.498	Nuclear
*LuGRF2*	L.us.o.m.scaffold210.25	Chr01:22366715-22366958(−)	496	54,341.46	8.94	50.02	61.79	−0.661	Nuclear
*LuGRF3*	L.us.o.m.scaffold55.291	Chr02:1276876-1277354(+)	232	25,895.75	8.92	56.31	49.57	−0.741	Nuclear
*LuGRF4*	L.us.o.m.scaffold248.2	Chr02:2602840-2602896(+)	387	43,513.62	8.96	77.55	51.47	−0.942	Nuclear
*LuGRF5*	L.us.o.m.scaffold38.164	Chr02:3705117-3705735(−)	439	48,533.09	6.31	58.68	59.38	−0.695	Nuclear
*LuGRF6*	L.us.o.m.scaffold3.735	Chr04:4408868-4409294(−)	369	40,068.02	6.50	51.32	57.51	−0.710	Nuclear
*LuGRF7*	L.us.o.m.scaffold77.97	Chr05:963201-963414(+)	505	55,704.35	9.90	68.90	54.67	−0.729	Nuclear
*LuGRF8*	L.us.o.m.scaffold15.308	Chr05:19180808-19180877(+)	151	17,727.24	9.51	68.70	72.85	−0.871	Nuclear
*LuGRF9*	L.us.o.m.scaffold102.125	Chr06:503595-503793(+)	159	17,644.74	9.22	55.41	51.51	−0.695	Nuclear
*LuGRF10*	L.us.o.m.scaffold148.58	Chr07:8111972-8112233(+)	474	51,511.23	8.48	50.30	60.95	−0.630	Nuclear
*LuGRF11*	L.us.o.m.scaffold148.59	Chr07:8125492-8125726(−)	496	52,555.95	8.03	61.70	59.07	−0.514	Nuclear
*LuGRF12*	L.us.o.m.scaffold20.42	Chr09:12134586-12135067(−)	236	26,242.21	9.10	56.40	53.69	−0.694	Nuclear
*LuGRF13*	L.us.o.m.scaffold9.371	Chr10:2577384-2577494(+)	468	52,014.37	7.70	58.79	62.35	−0.640	Nuclear
*LuGRF14*	L.us.o.m.scaffold9.76	Chr10:4513900-4513906(−)	405	45,647.92	8.95	76.02	52.79	−0.974	Nuclear
*LuGRF15*	L.us.o.m.scaffold5.65	Chr11:7541113-7541311(+)	363	41,009.42	8.88	67.48	52.59	−0.911	Nuclear
*LuGRF16*	L.us.o.m.scaffold76.290	Chr11:15704503-15704602(+)	184	21,144.89	9.22	59.53	75.16	−0.791	Nuclear
*LuGRF17*	L.us.o.m.scaffold64.61	Chr14:14156499-14156608(−)	229	26,083.73	9.04	55.43	75.02	−0.568	Nuclear

**Table 2 ijms-24-17107-t002:** Potential miRNA targets of *LuGRF* gene.

miRNA	Target	Expectation	miRNA Length	Target_start	Target_end	Inhibition	Multiplicity
lus-miR390a/b/c/d	*LuGRF7*	3	21	133	153	Cleavage	1
lus-miR390a/b/c/d	*LuGRF9*	3	21	118	138	Cleavage	1
lus-miR396a/b/c/e	*LuGRF1*	3	21	612	633	Cleavage	1
lus-miR396a/b/c/e	*LuGRF5*	3	21	546	567	Cleavage	1
lus-miR396a/b/c/e	*LuGRF4*	3	21	381	402	Cleavage	1
lus-miR396a/b/c/e	*LuGRF13*	3	21	554	575	Cleavage	1
lus-miR396a/b/c/e	*LuGRF14*	3	21	378	399	Cleavage	1
lus-miR396a/b/c/e	*LuGRF7*	3	21	357	378	Cleavage	1
lus-miR396a/b/c/e	*LuGRF9*	3	21	342	363	Cleavage	1
lus-miR396a/b/c/e	*LuGRF15*	3	21	360	381	Cleavage	1
lus-miR396a/b/c/e	*LuGRF8*	3	21	423	444	Cleavage	2
lus-miR396a/b/c/e	*LuGRF16*	3	21	531	552	Cleavage	1
lus-miR396a/b/c/e	*LuGRF11*	3	21	615	636	Cleavage	1
lus-miR396a/b/c/e	*LuGRF6*	3	21	540	561	Cleavage	1
lus-miR396a/b/c/e	*LuGRF10*	3	21	633	654	Cleavage	1
lus-miR396a/b/c/e	*LuGRF2*	3	21	615	636	Cleavage	1
lus-miR396a/b/c/e	*LuGRF17*	3	21	537	558	Cleavage	1
lus-miR396a/b/c/e	*LuGRF8*	3.5	21	286	306	Cleavage	2
lus-miR164a/b/c/d/e	*LuGRF1*	4	21	176	196	Translation	1
lus-miR164a/b/c/d/e	*LuGRF11*	4	21	179	199	Translation	1
lus-miR172a/b/c/d/e/f/g/h/i	*LuGRF4*	4	21	877	897	Cleavage	1
lus-miR172a/b/c/d/e/f/g/h/i	*LuGRF14*	4.5	21	898	918	Cleavage	1
lus-miR159a	*LuGRF14*	5	21	24	44	Cleavage	1
lus-miR159a	*LuGRF4*	5	21	24	44	Cleavage	1
lus-miR168a/b	*LuGRF6*	5	21	1046	1066	Cleavage	1

**Table 3 ijms-24-17107-t003:** Two-dimensional structures of LuGRF proteins.

Gene Name	Sequence Length	Alpha Helix (Hh)	Extended Strand (Ee)	Random Coil (Cc)
*LuGRF1*	492	93/18.90%	78/15.85%	321/65.24%
*LuGRF2*	495	98/19.80%	88/17.78%	309/62.42%
*LuGRF3*	232	30/12.93%	54/23.28%	148/63.79%
*LuGRF4*	387	71/18.35%	66/17.05%	250/64.60%
*LuGRF5*	439	57/12.98%	86/19.59%	296/67.43%
*LuGRF6*	369	85/23.04%	67/18.16%	217/58.81%
*LuGRF7*	505	77/15.25%	106/20.99%	322/63.76%
*LuGRF8*	151	40/26.49%	23/15.23%	88/58.28%
*LuGRF9*	333	41/12.31%	56/16.82%	236/70.87%
*LuGRF10*	474	64/13.50%	89/18.78%	321/67.72%
*LuGRF11*	496	85/17.14%	75/15.12%	336/67.74%
*LuGRF12*	236	24/10.17%	59/25.00%	153/64.83%
*LuGRF13*	468	44/9.40%	114/24.36%	310/66.24%
*LuGRF14*	405	80/19.75%	78/19.26%	247/60.99%
*LuGRF15*	363	33/9.09%	76/20.94%	254/69.97%
*LuGRF16*	184	36/19.57%	33/17.93%	115/62.50%
*LuGRF17*	229	71/31.00%	50/21.83%	108/47.16%

## Data Availability

All data are reported in the article and Appendix A.

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
