# Peer review of "Genome-Wide Analysis of Flax (Linum usitatissimum L.) Growth-Regulating Factor (GRF) Transcription Factors"

_ijms, 2023, doi:10.3390/ijms242317107_

Round 1

Reviewer 1 Report

Comments and Suggestions for Authors

In the current work, the authors investigated the GRF transcription factors Flax (Linum Usitatissimum L.). The gene structure, cis-acting elements, miRNA target prediction, and subcellular localization assays were done in present work. Besides, the authors studied the gene expression of GRF genes in different organs and against hormones treatment at different time points. In my opinion, this manuscript has a good potential for publishing. The novelty of work is high and results cover the objectives. However, it should be revised, accordingly:

-          English of text needs to polish

-          Line 49: add space between genus and species of “Oryzasativa”

-          Line 51: Delete “in” from “in Arabidopsis thaliana”

-          Line 78: scientific name should be provided in italic format.

-          Add it to line 106: The instability index ranges from 50.02 to 77.55, indicating that GRFs are probably the unstable proteins.

-          In Table 1, why use “Average” for showing MW of protein?

-          Line 112: Delete “MAGA11.0”

-          Line 122: Arabidopsis must be written in italic, please apply in whole text.

-          Line 227: protein name should not be provided in italic.

-          Line 285: this sentence needs to rewrite.

-          I suggest adding these notes to line 405: “Moreover, putative cis-regulatory elements associated with hormones-responsive, such as ABA, and MeJA were recognized in promoter region of LuGRF genes. In plant species, hormones such as ABA, and MeJA are more linked with stress cell signaling pathways (Ku et al, 2018; Heidari et al, 2021). In addition, several transcription factors and ion transporter genes are induced by ABA and MeJA signaling in response to environment stimuli (Hashemipetroudi et al, 2023; Faraji et al, 2021). Our results suggest that LuGRF genes might be   directly induced by ABA and MeJA.”

-          References:

Ku et al, 2018: https://doi.org/10.3390/ijms19103206

Heidari et al, 2021: https://doi.org/10.3390/agronomy11061146

Hashemipetroudi et al, 2023: https://www.frontiersin.org/articles/10.3389/fpls.2023.1112354/full

Faraji et al, 2021: https://doi.org/10.1007/s10534-021-00301-4

-          Lines 422-425: these lines should be edited and be improved.

-          Line 428: Please correct: “Using ExPASyProtParam LuGRF gene protein database”. The ProtParam tool is just acceptable.

-          Line 434: Phylogenetic analysis section is not complete.  Add more details related to multiple alignment etc.

-          Lines 454-455:  these lines should be edited and be improved.

-          Line 468: agrobacterium tumeficans be modified to Agrobacterium tumeficans and provide in italic

-          Lines 483-485: Add references for all tools and packages were used.

-          Line 498: “17 LuGRF gene families” be modified to “17 LuGRF family members”

-          Conclusion is not acceptable. It is so short without any real conclusion.

-          Figures are well illustrated.

Comments on the Quality of English Language

Please follow my comments and suggestions for Authors.

Reviewer 2 Report

Comments and Suggestions for Authors

This paper deals with the investigation of the genes encoding GRF transcription factor in flax. Although flax is an interesting crop, it seems that the research design in this paper is not good. I would like to know the results of the transcriptome analysis, but it is difficult for me to understand the data deposited in the NCBI database.

In Figure 7, RFP images are incorrect. They are "Merged" images.

Comments on the Quality of English Language

I think editing of English language is required.

For example,

Line 57-58, "Previous studies have found that GRF gene is 57 widely involved in the growth and development, it was reported to respond with stress 58 of various plant tissues and organs [16-17]."

Line 76-78, "If AtGRF7 can inhibit DEHYDRATION response 76 element binding protein encoding gene expression of DREB2A to regulate plant response 77 to osmotic stress [2]."

Reviewer 3 Report

Comments and Suggestions for Authors

The authors conducted a bioinformatics analysis, identified 17 Flax GRF transcription factors, and provided more information on their gene structure, promoter motifs, sub-cellular location, etc.

Although the work is a useful catalog for further improving the knowledge of GRF family genes, the manuscript needs some revisions.

The manuscript could be better written, especially in the materials and method section, with many descriptions that look jumbled here and there. For instance, lines 449-451. Indicate the developmental stage and other details whenever explaining the results. Line 266-272: Clearly state the type of treatments and samples.

The figures 5 and 6 may be moved to supplementary information as these are not adding anything significantly beyond the description.

Others

Line 49: "Oryzasativa"

Line 76-78: Sentence structure

Comments on the Quality of English Language

The authors are recommended to improve the manuscript with professional English editing service

Reviewer 4 Report

Comments and Suggestions for Authors

In my opinion, the manuscript meets the requirements for publication in IJMS. However, it requires a few minor corrections.

1. Abstract, lines 18-19: Please specify that this was a bioinformatics analysis and that you used a flax DNA sequence available in the NCBI database.

2. Abstract, line 20: "and their encoded proteins ALL..." - this fragment is misleading because earlier in this fragment there is information about 15 genes out of the 17 identified.

3. Abstract, line 21: "51 GRF genes" - it is worth adding that these are genes from other species, because without this information it is misleading, considering the previous information about 17 GRF genes in flax.

4. Results, lines 118-123: Please verify that the information provided is correct. According to Figure 1, there are 4 genes in subfamilies I-III and 5 genes in subfamily IV. I am also wondering about the use of the word "clusters" - wouldn't it be better just to call them "genes"?

5. Results, line 243 and Figure 8B: The text contains information about the lack of LuGRF gene expression in the root, stem, and shoot, while Figure 8B shows differential expression in these tissues in the control variants. Where do these discrepancies come from?

6. Discussion: I am missing a reference to the fact that the structure of GRF proteins is so different (motifs in Figure 2b) - do proteins with different structures have similar functions? Do the authors' results and previous publications indicate any connection between structure and function? In particular, proteins 4, 14, and 15 stand out from the rest.

Minor comments:

1. Title: GRF --> (GRF).

2. Line 76: Remove "If".

3. Please write the species names in italics throughout the manuscript and capitalize only the first part of the name.

4. Figure 2: What do CDS and UTR stand for?

5. Line 160: "WGD" - explain the abbreviation.

6. Figure 5: In my opinion, it can be moved to supplementary materials.

7. Figures 8B and 8C: They would be clearer if the order of the LuGRF genes were the same.

8. Discussion, lines 337-338: The sentence is unclear.

9. Materials and methods, line 502: The sentence is unclear.

Comments on the Quality of English Language

The English language is understandable, only the abstract and partially the materials and methods (especially lines 482-486) require some adjustments to make them clear.

Round 2

Reviewer 1 Report

Comments and Suggestions for Authors

The new version is improved. In my opinion, it can be accepted.

Reviewer 2 Report

Comments and Suggestions for Authors

I understand the modified results of the transcriptome database analysis. I noticed another point. In Materials and Methods section, line 438, the sub-title includes "GO enrichment analysis" but it seems that the explanation is not described sufficiently. 
